Future habitat changes of Bactrocera minax Enderlein along the Yangtze River Basin using the optimal MaxEnt model

Fu Chun 1
Wang Xian 2
Huang Tingting 3
Wang Rulin 4 5 wrl_1986_1@163.com
1 Key Laboratory of Sichuan Province for Bamboo Pests Control and Resource Development, Leshan Normal University , Leshan , China
2 Hejiang Bureau of Agriculture and Rural Affairs , Hejiang , China
3 Chengdu Agricultural Technology Extension Station , Chengdu, Sichuan , China
4 Sichuan Provincial Rural Economic Information Center , Chengdu , China
5 Water-Saving Agriculture in Southern Hill Area Key Laboratory of Sichuan Province , Chengdu, Sichuan , China
Negri Ilaria
Electronic publication date: 2023 Nov 21
Publication date: 2023
Volume: 11
Electronic Location ID: e16459
Received 2023 Jun 7; Accepted 2023 Oct 23
Copyright: © 2023 Fu et al.
Copyright year: 2023
Copyright holder: Fu et al.
License: This is an open access article distributed under the terms of the Creative Commons Attribution License, which permits unrestricted use, distribution, reproduction and adaptation in any medium and for any purpose provided that it is properly attributed. For attribution, the original author(s), title, publication source (PeerJ) and either DOI or URL of the article must be cited.
License URL: https://creativecommons.org/licenses/by/4.0/

Keywords: Bactrocera minax, Climate change, The optimal MaxEnt model, Suitable habitat, The Yangtze River Basin

Funding: Science and Technology Project of Leshan Normal University LZD010, XJR17005, 2022SSDJS005 and KYPY2023-0006 This research was funded by the Science and Technology Project of Leshan Normal University (LZD010, XJR17005, 2022SSDJS005, KYPY2023-0006). The funders had no role in study design, data collection and analysis, decision to publish, or preparation of the manuscript.

==============================
Background

Bactrocera minax (Enderlein, 1920) (Diptera: Tephritidae) is a destructive citrus pest. It is mainly distributed throughout Shaanxi, Sichuan, Chongqing, Guizhou, Yunnan, Hubei, Hunan, and Guangxi in China and is considered to be a second-class pest that is prohibited from entering that country. Climate change, new farming techniques, and increased international trade has caused the habitable area of this pest to gradually expand. Understanding the suitable habitats of B. minax under future climate scenarios may be crucial to reveal the expansion pattern of the insect and develop corresponding prevention strategies in China.

Methods

Using on the current 199 distribution points and 11 environmental variables for B. minax, we chose the optimal MaxEnt model to screen the dominant factors that affect the distribution of B. minax and to predict the potential future distribution of B. minax in China under two shared socio-economic pathways (SSP1-2.6, SSP5-8.5).

Results

The current habitat of B. minax is located at 24.1–34.6°N and 101.1–122.9°E, which encompasses the provinces of Guizhou, Sichuan, Hubei, Hunan, Chongqing, and Yunnan (21.64 × 104 km2). Under future climate scenarios, the potential suitable habitat for B. minax may expand significantly toward the lower-middle reaches of the Yangtze River. The land coverage of highly suitable habitats may increase from 21.64 × 104 km2 to 26.35 × 104 × 104 km2 (2050s, SSP5-8.5) ~ 33.51 × 104 km2 (2090s, SSP5-8.5). This expansion area accounts for 29% (2050s, SSP1-2.6) to 34.83% (2090s, SSP1-2.6) of the current habitat. The center of the suitable habitat was predicted to expand towards the northeast, and the scenario with a stronger radiative force corresponded to a more marked movement of the center toward higher latitudes. A jackknife test showed that the dominant variables affecting the distribution of B. minax were the mean temperature of the driest quarter (bio9), the annual precipitation (bio12), the mean diurnal range (bio2), the temperature annual range (bio7), and the altitude (alt).

Discussion

Currently, it is possible for B. minax to expand its damaging presence. Regions with appropriate climate conditions and distribution of host plants may become potential habitats for the insects, and local authorities should strengthen their detection and prevention strategies. Climate changes in the future may promote the survival and expansion of B. minax species in China, which is represented by the significant increase of suitable habitats toward regions of high altitudes and latitudes across all directions but with some shrinkage in the east and west sides.

Introduction

Bactrocera minax (Enderlein, 1920) (Diptera: Tephritidae) is a destructive citrus pest (Liu et al., 2020). It is mainly distributed throughout China, Bhutan, India, Sikkim, and Japan and is an A1 quarantine pest according to the European and Mediterranean Plant Protection Organization (Xu et al., 2017). In China, it is mainly found in Shaanxi, Sichuan, Chongqing, Guizhou, Yunnan, Hubei, Hunan, and Guangxi, and is a second-class pest that is prohibited from entering the country (Xu et al., 2017). It is univoltine and overwinters in the soil as pupae because high soil temperatures before eclosion are harmful to the emergence of B. minax, and may cause death (Li et al., 2012; Ma et al., 2017). With climate changes, the adjustment of planting techniques, and the increase of international trade, areas harmful to the species are gradually expanding. The suitable temperature for adult activity is 20–23 °C, and the life span is shortened when the temperature exceeds 30 °C (Ma et al., 2017). The pest mainly harms citrus plants such as lime, lemon, grapefruit, Chinese wolfberry, sweet orange, bergamot, red orange, and grapefruit (Liu et al., 2014). Female adult insects lay their eggs in the pericarp, and the larvae feed on the sarcocarp. Consequently, the affected fruits may appear reddish yellow and even fall off the trees while not yet ripe, leading to harvest losses (and eventually economic losses for the citrus farmers) of more than 50% or even 100%. This insect is considered the top pest for citrus production in China (Zhang et al., 2015).

Temperature rises may affect the growth and development behavior of individual organisms and may also affect the balance of competition, predation, and parasitism. Imbalances may seriously impact biodiversity and ecosystem functions (Sillett et al., 2022; Zhang et al., 2021; Henry et al., 2017). As ectotherms, insects are very sensitive to changes in environmental temperature (Mukherjee et al., 2023). The change of environmental temperature will inevitably affect the growth, development, survival, reproduction, and diffusion of insects (Iltis et al., 2019; Hou et al., 2022). According to the law of effective accumulative temperature, as the temperature increases within a certain temperature range, the growth and development speed of pests accelerate, and the amount of time that the insect is able to be destructive is increased (Arora & Srivastava, 2021; Hou et al., 2022). In order to adapt to global warming, insects will migrate and spread to high altitude and high latitude areas (Ji et al., 2020; Fois et al., 2018). Huang, Jiang & Li (2020) showed that under the influence of climate change, variations in farming systems, and replacement varieties of certain crops, there has been a change to the species of wheat pests and diseases; aphids have risen to the most important pests in the Huang-Huai-Hai wheat region and the occurrence area has experienced an expansion to the north. Cai et al. (2022) showed through regression analysis that with the increase of annual average temperature in China, the melon fruit fly’s first occurrence became significant delayed with an earlier population peak stage. Yang et al. (2013) found that the scope of areas damaged by B. minax has increased from 128 cities in seven provinces and regions in 1995 to 167 cities in nine provinces and regions in 2010. As climate warming continues, understanding the suitable habitats of B. minax in China under future climate scenarios may be crucial to determine the expansion law of the insect and to develop corresponding prevention strategies.

Ecological niche models (ENMs) can combine the occurrence record of species with environmental factors to simulate the niche and potential geographical distribution of species in the natural environment (Carrillo-García et al., 2023; Banerjee et al., 2019). In recent years, the application of ENMs in the prediction of the potential geographical distribution of quarantine pests has become increasingly widespread. Zhou et al. (2022) used Climex and MaxEnt to predict the global distribution of the citrus quarantine pest Anoplophora chinensis, emphasizing the need to strengthen the control, monitoring, and quarantine of the pest in the threatened areas. Wang et al. (2020) analyzed the geographical distribution and changes of the international quarantine pest of citrus Diaphorina citri in China under the background of climate change, providing a reference for the formulation of control measures in different provinces and regions. Wang et al. (2023a) analyzed the invasion risk of three scarab beetles (Popillia japonica, Amphimallon majale, and Heteronychus arato) using the MaxEnt model, suggesting that the local agriculture, forestry, and customs departments in China should pay more attention to the presence of this pest. Tang et al. (2019) predicted the suitable area of Euplatypus parallelus in China by the Maxent model, and the result showed that the pest has invaded Hainan and Taiwan. Compared to CLIMEX and GARP, MaxEnt takes the influence of climate factors on the distribution area of a species into consideration (Zhang et al., 2016). Non-climate factors such as elevation, landscape factors, and land-use type are also addressed in MaxEnt, making the model produce the results closest to reality (Munro et al., 2022; Koide & Kadoya, 2019).

Only a few articles have investigated some Tephritidae insects for their future potential distribution in China (Qin et al., 2019; Wang et al., 2022; Zhang, Zhao & Li, 2021; Sun et al., 2017). However, no reports have predicted the distribution of B. minax under future climate change scenarios. In this study, we implemented the MaxEnt model to screen the critical environmental factors that affect the distribution of B. minax based on the distribution data of B. minax and climate data. We also predicted the potential suitable habitat for B. minax in China and its alteration trends under future climate change scenarios. We hope to better understand the insect’s expansion risks and develop prevention strategies against its spread.

Materials and Methods

Study area

Originating from the eastern Qinghai-Tibet Plateau, the Yangtze River flows eastward into the Chinese East Sea with a total length of ~6,300 km, making it the longest river in China and the third longest river in the world (Boscari et al., 2022; Li, Zeng & Long, 2020). The Yangtze River basin (90°33′E~2°25′E, 24°30′N~35°45′N) covers an area of 1.8 million km2 that is composed of numerous terrain types including plateaus (Qinghai-Tibet and Yunnan-Guizhou), mountain ranges (Hengduan), basins (Sichuan), hilly regions (Chiang-nan), and plains in the lower-middle reaches of the river (Shi et al., 2022; Li et al., 2021; Liu et al., 2018). The river basin is dominated by the subtropical monsoon climate except for the Qinghai-Tibet Plateau section (highland alpine climate). The annual temperature is generally higher in the eastern and southern parts of the river basin. The annual precipitation in the basin is about 1,100 mm, and 70–90% of this falls from May to October. Since citrus cannot grow on the Qinghai-Tibet Plateau and no pest damages have been recorded, this area is excluded from the study area (Zhai et al., 2022; Song et al., 2020) (Fig. 1).

Figure 1 A representation of the study area and points of B. minax sample records.

The boundary map was obtained free from Natural Earth (http://www.naturalearthdata.com/). Based on the principle of national and territorial integrity, we have modified and adjusted the vector boundary.

Species occurrence and environmental data

The occurrence of B. minax was determined via field survey and literature search. The geographical distribution of B. minax was obtained during the field surveys. In the literature search, published articles related to B. minax and the species distribution database were searched to accurately simulate the distribution of the pest. Exact locations in the acquired distribution data were adopted directly if expressed in latitudes and longitudes; otherwise, the coordinates were found according to the mentioned townships or villages (Wang et al., 2020; Yang et al., 2022). ENMTOOLs was used to calculate the distance between the grid center and each point, and only the distribution record with the smallest distance was maintained (Liu et al., 2021). In the end, 199 distribution points were recorded.

A total of 19 bioclimatic variables were calculated according to the daily observation data provided by the meteorological stations in China from 2001 to 2020. The surface climate data from the China Meteorological Data Sharing Service System were used after excluding the daily average temperature and the missing measurement value of precipitation at 20–20 h, and converting snow, ice and fog into precipitation (Mao et al., 2022; Zhao et al., 2022). Future climatic data were obtained from the WorldClim data set (https://www.worldclim.org/) with a spatial resolution of 2.5 arc-minutes and a temporal scale of a month. Twenty variables, including the altitude (alt) and 19 bioclimatic variables are shown in Table S1. The data periods used in this study include the current period (2000–2020), the 2050s (2041–2060), and the 2090s (2081–2100).

We chose Climate Model Intercomparison Project 6 (CMIP6) as the climate prediction mode. Future bioclimatic variables were obtained from the WorldClim data set (https://www.worldclim.org/) with a spatial resolution of 2.5 arc-minutes. Compared to CMIP5, CMIP6 show the effects of aerosols on short-term climate changes, thereby being able to better represent the impact of the changes in land-use patterns on regional climates and giving more accurate predictions (Niu et al., 2023; Liu et al., 2022). The climate change scenarios SSP1-2.6 (sustainable development with low radiative forcing), SSP2-4.5 (intermediate development with intermediate radiative forcing), and SSP5-8.5 (normal development with high radiative forcing) under the BCC-CSM2-MR mode were used in this study (Kajtar et al., 2022; Huntingford, Williamson & Nijsse, 2020; Seneviratne & Hauser, 2020; Carvalho et al., 2022).

Selection of environmental variables

Due to the multicollinearity between bioclimatic variables, screening should be conducted before constructing the model (Wei et al., 2019; Wang et al., 2019). As a result, the parameter estimation variance would rise and the simulation results may be affected if variables with high intercorrelations and low contribution rates are not excluded (Wang et al., 2020; Wang et al., 2023b; Tang et al., 2019). In this study, 20 environmental variables and 199 recorded distribution points were used to construct an initial MaxEnt model for calculating percent contribution rate of variables (Wang et al., 2023b; Xia et al., 2023). Variables with a contribution rate smaller than 0.1 were excluded. The remaining variables were assigned pairwise for the Pearson correlation analysis of multicollinearity by the SPSS software (Fig. 2). If two environmental variables had a Pearson coefficient |r| ≥ 0.8, colinearity between the two was confirmed, and the one with a higher contribution rate would be preserved for subsequent model analysis. Otherwise, both variables must be preserved since no colinearity was present (Yang et al., 2022; Guan et al., 2022). In the end, alt (altitude), mean diurnal range (bio2), isothermality (bio3), temperature annual range (bio7), mean temperature of driest quarter (bio9), mean temperature of warmest quarter (bio10), annual precipitation (bio12), precipitation of wettest month (bio13), precipitation of driest month (bio14), precipitation seasonality (bio15), and precipitation of warmest quarter (bio18) were selected to construct the final MaxEnt model.

Figure 2 The matrix of Pearson correlation coefficients.

Modelling progress of MaxEnt

The operational procedure for the MaxEnt software (Version 3.4.4, developed by (Phillips, Anderson & Schapire, 2006)) was as follows: (1) Occurrence of the species B. minax in “CSV” format and the environmental variable in “ASC” format were imported into the “sample” and “environmental layers” data boxes, respectively. (2) The option called “Do jackknife to measure variable importance” was selected respectively to measure the importance of variables. (3) In the initial model, “random test percentage” was set to 25 %, while in the reconstructed model, “random seed” was selected, and the “replicates” was set to 10 (Bao, Li & Zheng, 2022; Dou et al., 2022; Rhoden, Peterman & Taylor, 2017; Xia et al., 2023; Wang et al., 2023b). (4) The Kuenm software package of R language was used to optimize the regularization multiplier (40, with values of 0.1~40) and feature combination (L, Q, P, T, H, LQ, LP, LT, LH, QP, QT, QH, PT, PH, TH, LQP, LQT, LQH, LPT, LPH, QPT, QPT, QPH, QTH, QTH, LQPT, LQPH, LQPH, LQTH, LQTH and LQPTH) of the model, and the optimal setting of the minimum information criterion AICc value (delta.AICc) among 1,160 results was selected (Zhao et al., 2022; Xia et al., 2023; Wang et al., 2023b).

With the constructed MaxEnt model, the potential distribution possibility P (ranging between 0-1) of B. minax across China was obtained using the Reclassify function in ArcGIS. Based on the P values and the occurrence of B. minax damages in reality, regions were recognized as non-habitats (no-risk area, 0 ≤ P < 0.05), poorly suitable habitats (low-risk area, 0.05 ≤ P < 0.33), moderately suitable habitats (moderate-risk area, 0.33 ≤ P < 0.66), and highly suitable habitats (high-risk area, P ≥ 0.66) (Wang et al., 2020; Arora & Srivastava, 2021; Wang et al., 2023b). The spatial analysis tool in ArcGIS was used to calculate the area of suitable habitats, and the distribution changes between binary SDMs tool was selected to calculate and visualize the coordinates of the centroid (Qin et al., 2019; Tang et al., 2017). The distance and direction of centroid migration were calculated using Yue et al. (2011)’s method.

In order to determine model accuracy, we selected the area under the receiver operating characteristic curve (AUC) and true skill statistics (TSS) as evaluation indicators. The range of AUC values was (0, 1), and a larger value indicates a higher model accuracy (Sreekumar & Nameer, 2022). The range of TSS values was (−1, 1); the closer the value is to 1, the better the predictive performance of the model (Allouche, Tsoar & Kadmon, 2010).

Results

Evaluation of the MaxEnt model accuracy

As can be inferred from Table 1, the AUC values of training data and test data under the current conditions were 0.927 ± 0.006 and 0.917 ± 0.044, respectively. Under the future climate scenario models, the AUC of training data and test data varied between 0.925 ± 0.004 to 0.932 ± 012.002 and 0.918 ± 0.004 to 0.924 ± 0.035, respectively. The TSS value under the current condition was 0.879 ± 0.014. Under the future climate scenarios, the TSS values ranged from 0.872 ± 0.098 to 0.909 ± 0.042.

Table 1 AUC and TSS values of all the models.

Years	Scenario	AUC	TSS	
Training data	Test data	
Current	/	0.927 ± 0.006	0.917 ± 0.044	0.879 ± 0.014	
2050s	SSP1-2.6	0.93 ± 0.005	0.921 ± 0.043	0.872 ± 0.098	
2090s	SSP5-8.5	0.931 ± 0.003	0.924 ± 0.035	0.898 ± 0.032	
SSP1-2.6	0.925 ± 0.004	0.918 ± 0.004	0.883 ± 0.014	
SSP5-8.5	0.932 ± 0.002	0.919 ± 0.028	0.909 ± 0.042	

Potential habitats of B. minax under different climates

Figure 3A shows the habitats of B. minax in the Yangtze River Basin under current climate conditions. The highly suitable habitats can be divided into two regions, one located in middle-eastern Sichuan and western Chongqing in the upper reach and another expanding from eastern Guizhou toward western Hubei (in the middle reach) via northwestern Hunan. The total land area of the highly suitable habitats was 21.64 × 104 km2, accounting for 29.82% of all the suitable habitats. The moderately suitable habitats for this pest are scattered throughout Sichuan, Chongqing, Guizhou, and Hubei, covering an area of 19.54 × 104 km2. The poorly suitable habitats stretch from the Sichuan basin in the upper reach toward the lower reach of the Yangtze River, covering an area of 31.38 × 104 km2.

Figure 3 Potential habitats of B. minax under climate change scenarios.

(A) Current climate conditions (1991–2020); (B) 2050s, SSP1-2.6; (C) 2050s, SSP5-8.5; (D) 2090s, SSP1-2.6; (E) 2090s, SSP5-8.5; (F) statistical analysis for different habitat types under different times and climate scenarios. Based on the principle of national and territorial integrity, we have modified and adjusted the vector boundary. Boundary map source: Natural Earth (http://www.naturalearthdata.com/).

Figures 3B–3E depicted the suitable habitats of B. minax at different periods (2050s, 2090s) and climate scenarios (SSP1-2.6, SSP5-8.5). Statistical analysis revealed that the land areas of highly suitable habitats, moderately suitable habitats, and total suitable habitats would rise significantly (Fig. 3F). More specifically, for highly suitable habitats, the land coverage increased from 21.64 × 104 km2 to 26.35 × 104 km2 (2050s, SSP5-8.5) ~ 33.51 × 104 km2 (2090s, SSP5-8.5). For moderately suitable habitats, the variations ranged from 19.54 × 104 km2 to 27.07 × 104 km2 (2050s, SSP1-2.6) ~ 32.71 × 104 km2 (2090s, SSP5-8.5). The total suitable habitat of B. minax ranged from 72.57 × 104 km2 to 94.54 × 104 km2 (2050s, SSP1-2.6) ~ 103.96 × 104 km2 (2050s, SSP5-8.5).

Changes in the potential suitable habitats of B. minax in China under future climate scenarios

We performed overlay calculations on the figures showing future and current predictions in ArcGIS to provide a more intuitive representation of the future changes in suitable habitats (Fig. 4). The stable area, defined as the B. minax habitat both now and in the future, accounts for 93.5% (2090s, SSP5-8.5) to 95.34% (2050s, SSP5-8.5) of the current habitat. The shrinkage area, defined as the current but not future B. minax habitat, accounts for 4.66% (2050s, SSP5-8.5) to 6.5% (2090s, SSP5-8.5) of the current habitat. The expansion area, defined as the B. minax future but not current habitat, accounts for 29% (2050s, SSP1-2.6) ~ 34.83% (2090s, SSP1-2.6) of the current habitats. Changes in the suitable habitats in the 2050s and 2090s (Figs. 4G and 4H, respectively) were obtained by the overlay analysis function in ArcGIS. New habitats may be seen in all reaches of the Yangtze River, and the shrinkage area, seen as scattered small regions in all reaches of the river, accounts for a low ratio in all the areas.

Figure 4 Changes in the layout of suitable habitats of B. minax in China under future climate scenarios.

(A) 2050s, SSP1-2.6; (B) 2050s, SSP5-8.5; (C) 2050s; (D) 2090s, SSP1-2.6; (E) 2090s, SSP5-8.5; (F) 2090s. Based on the principle of national and territorial integrity, we have modified and adjusted the vector boundary. Boundary map source: Natural Earth (http://www.naturalearthdata.com/).

The center point of the suitable habitats of B. minax was calculated using the method proposed by Yue et al. (2011). Under the current climate conditions, the center point was located at 29.42°N, 109.74°E (Fig. 5). Under the SSP1-2.6 scenario, the point moved 46.4 km northeast to 29.49°N, 110.21°E by the 2050s and 54 km further northeast to 29.53°N, 110.74°E by the 2090s. Under the SSP5-8.5 scenario, the point moved 142.59 km northeast to 29.48°N, 111.17°E by 2050s and a further 116.36 km northwest to 29.61°N, 110.02°E by the 2090s.

Figure 5 The movement of the center point of potential B. minax suitable habitats in China under different environmental scenarios.

Based on the principle of national and territorial integrity, we have modified and adjusted the vector boundary. Boundary map source: Natural Earth (http://www.naturalearthdata.com/).

Dominant variables affecting the distribution of B. minax

The Jackknife test showed that, when modeling ‘with only variable’, the mean temperature of driest quarter (bio9), the annual precipitation (bio12), and the mean diurnal range (bio2) were the three variables with the highest regularized training gain, test gain, and AUC values, indicating that they contain more effective information (Fig. 6). When modelling ‘without variables’, the regularized training gain, test gain, and AUC value of the mean temperature of driest quarter (bio9), the temperature annual range (bio7) and the altitude (alt) showed the greatest decrease, indicating that they have a significant impact on the model results (Fig. 6). Overall, the dominant variables affecting the distribution of B. minax were the mean temperature of driest quarter (bio9), the annual precipitation (bio12), the mean diurnal range (bio2), the temperature annual range (bio7), and the altitude (alt).

Figure 6 Jackknife test of environmental variables.

Discussion

The Jackknife test showed that the mean temperature of driest quarter (bio9), the mean diurnal range (bio2) and the temperature annual range (bio7) are the key temperature factors affecting the distribution of B. minax. The driest quarter in China is generally winter and spring, which is consistent with the conclusion that the average temperature in winter and spring is a key climate factor affecting the occurrence and harm of B. minax discovered by Cai et al. (2022). Low temperatures in winter is the main factor affecting the growth and development of B. minax. When the temperature is short or continually at 0 °C, the insect can only partially pupate and emerge or not at all (Liu et al., 2013). B. minax development must go through a certain stage of low-temperature diapause (Wang & Zhang, 2009), which explains why the highly suitable habitats are mainly distributed in the cooler climate surrounding mountainous areas (middle-eastern Sichuan province, western Chongqing and eastern Guizhou province toward western Hubei). The temperature differences in different regions due to latitude, altitude, and even different years within the same region can directly affect the time of adult emergence (Wang & Zhang, 2009; Zhang et al., 2015), indicating that the mean diurnal range is an important indicator that affects the species’ suitable habitat.

The significant increase of annual average temperature is one of the most obvious characteristics of climate warming (Zhou et al., 2022; Zhang et al., 2021; Seneviratne & Hauser, 2020). Due to the impact of climate change, the growth and development rate of most pests are accelerated, and the damage period is advanced, which makes pest control more difficult (Bajwa et al., 2020; Xue et al., 2019). Studies have found that the current climate warming trend will aggravate the harm of phytophagous pests, promote their population growth and geographical expansion, increase the frequency of pest outbreaks and seriously threaten agricultural production safety (Jezeer et al., 2019; Guzmán et al., 2016). Individual specimens of B. minax have been found in areas with an altitude of 230–1,850 m and a latitude of 24–33°N, which is smaller than the range of the suitable habitat (24.1–34.6°N) predicted in this study (Zhang et al., 2015; Tang et al., 2012). Therefore, under current situation, there are space and possibilities for B. minax damages to expand. Appropriate actions must be taken in regions where B. minax damages have been detected to prevent its spread. Yang et al. (2013) used the CLIMEX model to analyze the suitability of B. minax survival across China and concluded that this insect shall not spread to areas north of 34.41°N, which is generally consistent with our finding. The significant differences in the habitats between previous studies and our may be explained by the varied prediction rationales of the models chosen.

Under the three future climate scenarios, the temperature and precipitation across China will rise with time, especially south of the Yangtze River (Shi et al., 2022; Li et al., 2021), which is beneficial for the overwintering, growth, and reproduction of B. minax. This may explain why the prediction results for the land areas of moderately and highly suitable habitats south of the Yangtze River in the 2090s are significantly larger than those in the 2050s. However, the land area of suitable habitats is not always positively correlated to temperature rise, and area shrinkage is observed for some poorly suitable habitats, especially under SSP5-8.5. Under this scenario, maximum shrinkage happens in the 2090s when numerous poorly suitable habitats become non-habitats. This effect may be explained by the temperature and precipitation exceeding the maximum threshold for B. minax growth. In summary, climate changes in the future may promote the survival and expansion of B. minax in China, which is represented by the significant increase of suitable habitats toward regions of high altitudes and latitudes across all directions but with some shrinkage in the east and west sides. This discovery is generally consistent with previous conclusions that species tend to migrate to habitats at high latitudes and altitudes in the context of global warming (Ji et al., 2020; Fois et al., 2018). Meanwhile, under future climate change scenarios, the center of B. minax habitats moves northeast toward higher latitudes, and the movement becomes more obvious as the radiative forcing rises. Such movement patterns are also reflected in investigation of Hyphantria cunea (Ji, Su & Yu, 2019), Loxostege sticticalis (Tang et al., 2017) and Spodoptera frugiperda (Jiang et al., 2022). In addition, the distribution of B. minax habitat largely depends on the geographical distribution of its host citrus. In the future, the climate of northern subtropical areas may become more appropriate for citrus growth, providing the basis for B. minax to spread north (Xian, Liu & Zhong, 2022). This diffusion trend has played a warning role in the prevention and control of citrus fruit flies, especially in the expand suitable habitats. Aside from common agricultural, chemical, radiative, and biological techniques, new technologies such as remote sensing, geographical informatics, and artificial intelligence may be integrated for this purpose as well (Li et al., 2019; Zhang et al., 2015).

Biological factors play a role in insect distribution, but so to do host, natural enemy, and self-diffusion capabilities (Faal & Teale, 2022). The B. minax adults’ abilities to fly and disperse are strong, and their flight distance can reach 500–1,500m (Yang et al., 2013). Although the host includes all species of citrus, B. minax prefers lime and sweet orange for feeding; adults also prefer navel orange for spawning (Wang & Zhang, 2009; Gong et al., 2020). However, non-biological factors cannot be quantified temporarily due to technical reasons, which may be the common dilemma of such research. Future research may provide prediction results that are closer to reality by integrating other models of ecological niches and remote sensing technologies and considering factors like the distribution of the host plant, terrain, and local intensity of damage prevention, so as to offer theoretical rationale and technical support for the detection and prevention of B. minax damages.

Supplemental Information

Supplemental Information 1 20 initial environmental variables used in this study.

Click here for additional data file.

Additional Information and Declarations

Competing Interests

Author Contributions

Data Availability

The authors declare that they have no competing interests.

Chun Fu conceived and designed the experiments, performed the experiments, prepared figures and/or tables, authored or reviewed drafts of the article, and approved the final draft.

Xian Wang performed the experiments, authored or reviewed drafts of the article, and approved the final draft.

Tingting Huang analyzed the data, prepared figures and/or tables, and approved the final draft.

Rulin Wang conceived and designed the experiments, performed the experiments, analyzed the data, prepared figures and/or tables, and approved the final draft.

The following information was supplied regarding data availability:

The data is available at figshare: Wang, Rulin (2023). Distribution data of Bactrocera minax. figshare. Dataset. https://doi.org/10.6084/m9.figshare.21928347.v1.

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
