# Peer review of "Future habitat changes of Bactrocera minax Enderlein along the Yangtze River Basin using the optimal MaxEnt model"

_PeerJ, doi:10.7717/peerj.16459_

## Round 0.1 · original submission · Minor Revisions

The aim of the study by Chun Fu and colleagues is to describe the actual distribution of Bactrocera minax, a destructive pest for citrus production in China, and to identify its potential geographical expansion along the Yangtze River basin under climate change scenarios (e.g., rise in temperature and precipitation which mostly affect regions south of the Yangtze River). This study is of great importance because it may help implement preventive and control measures for the management of this severe pest (which has also the potential to establish in citrus production areas outside of Asia). In my opinion, the manuscript needs only minor revisions prior to be published. In particular, some improvements are needed in the methodology (e.g., improving measures of accuracy) and some details should be also provided (e.g. on bioclimate variables).

Reviewer 1 ·

Basic reporting

no comment

Experimental design

no comment

Validity of the findings

no comment

Additional comments

Language of the manuscript and methodology need revision

Annotated reviews are not available for download in order to protect the identity of reviewers who chose to remain anonymous.

Reviewer 2 ·

Basic reporting

This manuscript (MS) examines the modelling and mapping of the current and future distribution of the fruit fly Bactrocera minax under climate change in China. MS's structure is appropriate and organized. However, I have some suggestions in analysis, model accuracy testing and data presentations for improvement. The entire MS required English editing. In some of the sentences, the connection is missing, and the authors have to take care of language and sentences.

Experimental design

Although the methods are adequately used for analysis and modelling, and authors used only AUC for determining model accuracy, some AUC is on criticism because it tends to exaggerate the model performance. Instead, True Skill Statistics (TSS) is more practical. Hence, TSS analysis is required and incorporated. Further Jackknife test need to be done to evaluate the relative importance of single explanatory variables in the Maxent model and output of Jackknife test required to be incorporated in the MS.

Validity of the findings

The model of the current study highlighted the potential area for B. minax expansion in China under Climate Changing Scenario. However, in the model accuracy assessment, some of the aspects need to be included and incorporatedd in the Manuscript. For the information, the output of optimization done under different combinations of regularization multipliers and feature classes, the relative contribution of important bioclimatic variables, Response curves of the critical environmental variables determining Bactrocera minax, and the distribution need to be incorporated in the Manuscript Presentation.
I suggest authors to provide occurrence data and details of bioclimate variables as supplementary information

Additional comments

1. Line 19-21: rewrite the sentences as "Bactrocera minax Enderlein is a serious pest of citrus in China. Under the changing climate, the modified planting methods and the increase of international trade, its suitable habitat area is gradually expanding.
2. Line 26: use the word ‘used’ rather than ‘implemented’
3. Line 35-36: rewrite the sentence predicted to move remarkably towards as ‘predicted to expand towards’
4. Line no 53-54: rewrite the sentence. Use the word ‘univoltine’rather once in a year
5. Line no 126: use ‘Species occurrence data’ in place of ‘distribution data’
6. Line number 129 : give references
7. Line no: 166-167: rewrite it

---

## Round 0.2 · Minor Revisions

All the comments made by the reviewers have been addressed but in my opinion, the manuscript still needs minor revisions, especially concerning the language and the clarity of the text.

Here are some suggestions:
In the Background, I suggest first of all to introduce the pest (e.g., XXX is a destructive pest of…). Also, I suggest removing details on the life cycle (univoltinism, overwintering stage) because they are not necessary.

Regarding the main text, here are some editing/suggestions, but the authors should carefully check the whole manuscript:
- please, change “occurs univoltine” to “is univoltine”.
- Please, change “overwinters in the soil with pupae” to “overwinters in the soil as pupae”.
- Please, change “The expansion area , accounts” to “The expansion area accounts”
- Please, change “the mean temperature of driest quarter (bio9), the annual precipitation (bio12) the mean diurnal range (bio2), the temperature annual range (bio7) and the altitude (alt)” into “the mean temperature of driest quarter (bio9), the annual precipitation (bio12), the mean diurnal range (bio2), the temperature annual range (bio7), and the altitude (alt)”.
- Please, change “As an ectotherm, insects are very sensitive to changes” to “As ectotherms, insects are very sensitive to changes”
- Please, change: “Species occurrence data and environmental data” to “Species occurrence and
- Please, change “The occurrence data of B. minax” to “Occurrence of the species B. minax”
- Please, change “This may explains” to “This may explain”

**Language Note:** The Academic Editor has identified that the English language must be improved. PeerJ can provide language editing services - please contact us at copyediting@peerj.com for pricing (be sure to provide your manuscript number and title). Alternatively, you should make your own arrangements to improve the language quality and provide details in your response letter. – PeerJ Staff

---

## Round 0.3 · accepted · Accept

Even if a few typos are still present (e.g. Climate change, new farming techniques, and increased international trade *has* caused; Using *on* the current 199 distribution points..), I think that the manuscript has been considerably improved, and is ready to be published.